# Ask4Help: Learning to Leverage an Expert for Embodied Tasks

**Kunal Pratap Singh** *
PRIOR, Allen Institute for AI

**Luca Weihs**
PRIOR, Allen Institute for AI

**Alvaro Herrasti**
PRIOR, Allen Institute for AI

**Jonghyun Choi**
Yonsei University

**Aniruddha Kembhavi**
PRIOR, Allen Institute for AI

**Roozbeh Mottaghi**
PRIOR, Allen Institute for AI

## Abstract

Embodied AI agents continue to become more capable every year with the advent of new models, environments, and benchmarks, but are still far away from being performant and reliable enough to be deployed in real, user-facing, applications. In this paper, we ask: *can we bridge this gap by enabling agents to ask for assistance from an expert such as a human being?* To this end, we propose the ASK4HELP policy that augments agents with the ability to request, and then use expert assistance. ASK4HELP policies can be efficiently trained without modifying the original agent's parameters and learn a desirable trade-off between task performance and the amount of requested help, thereby reducing the cost of querying the expert. We evaluate ASK4HELP on two different tasks – object goal navigation and room rearrangement and see substantial improvements in performance using minimal help. On object navigation, an agent that achieves a $52\%$ success rate is raised to $86\%$ with $13\%$ help and for rearrangement, the state-of-the-art model with a $7\%$ success rate is dramatically improved to $90.4\%$ using $39\%$ help. Human trials with ASK4HELP demonstrate the efficacy of our approach in practical scenarios.

## 1 Introduction

The journey toward creating multipurpose household assistants continues to pose many challenges despite recent progress in computer vision and robotics. Open source simulators [27, 32, 44] and benchmarks [45, 2] have enabled advancements in diverse tasks such as navigation [4, 21], interaction [15, 30], exploration [39, 9], instruction following [45, 2] and rearrangement [3, 52]. And yet, the best performing models (e.g., [25, 41]) on most tasks are not capable or reliable enough to be deployed in real-world applications.

For instance, the state-of-the-art (SoTA) in navigating towards a specified object category hovers around $50\%$[2] in the RoboTHOR [14] environment, and around $30\%$[3] in Habitat [32]. Slightly more complex tasks prove more daunting still, with the best models on Room Rearrangement [52] still below $10\%$[4]. In addition, these models often tend to get stuck, take repetitive actions and even collide

---

*correspond to kunals@allenai.org for any queries/comments

[2] https://leaderboard.allenai.org/robothor_objectnav/submissions/public

[3] https://aihabitat.org/challenge/2021/

[4] https://leaderboard.allenai.org/ithor_rearrangement_1phase/submissions/public

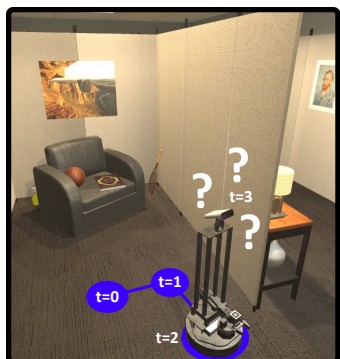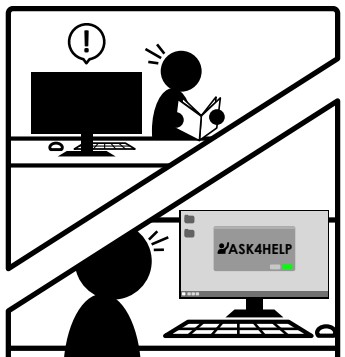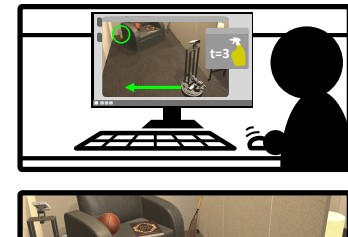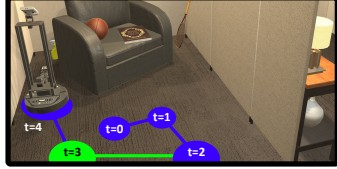

Figure 1: **Learning to ask for help** (ASK4HELP). *Left*: An agent struggles to find a spray bottle. *Middle*: The ASK4HELP policy recognizes this confusion and requests help from a human. *Right*: The human provides help at time $t = 3$, which puts the agent on the right path and it succeeds.

with furniture and walls as they move around in a scene. Deploying such agents in the real world can be costly, slow, and frustrating for users. In safety-critical situations such as rescue or reconnaissance operations, such failures can be even more expensive. While future modeling improvements may alleviate these concerns somewhat, we are still a ways away from creating agents that can perform household tasks robustly and safely.

Consider, for example, the setting of Fig. 1. A robot is deployed in a user's home and is tasked with locating objects, *e.g.* keys and remote controls, and is equipped with a SoTA navigation model such as EmbCLIP [25]. Even if we assume that this performance would seamlessly transfer from simulation to the real world, the agent would only be successful at its task about 50% [25] of the time. This is not good enough for deployment: few users would be satisfied with a household assistant which fails as frequently as it succeeds. If we want to deploy these models in practice we must improve their reliability. To this end, in this work, we assume access to an expert that can provide actions to our agent upon request but, additionally, we assume that querying this expert is costly and so the number of such queries should be minimized. Hence, we ask: *how can we dramatically improve agents' performance and reliability at tasks by using a small amount of expert assistance?*

Building embodied agents that can request and use help from humans presents new challenges. Firstly, how does an agent determine when it is most effective to ask for help? A hand-designed heuristic is likely to be sub-optimal from the standpoint of an agent's policy. Secondly, how can we find the right balance between ensuring that we only request the human to intervene when it is really necessary while maximizing the performance at the specified task? Lastly, how can we teach an agent to ask for help, without having to retrain the embodied agent that we are trying to support? This would avoid the need to have access to the entire model structure and weights, save a significant amount of compute, and enable wider applicability.

To this end, we propose the Ask For Help (ASK4HELP) policy which augments E-AI agents with the ability to request for help when necessary. Our ASK4HELP policy: (a) is minimally invasive – one can train this module without modifying the parameters of the underlying E-AI model; (b) can be trained efficiently, with a fraction of the training data, compute and time, compared to the original model; and (c) learns a desirable trade-off between task performance and the amount of help, and even presents a mechanism to the user to dynamically, at inference time without additional training, specify the cost of asking for help and thereby increase or decrease the amount of help that will be requested in future episodes.

We show the efficacy of our approach on the RoboTHOR object navigation [14] and AI2-THOR visual room rearrangement [52] tasks. We learn an ASK4HELP policy to support the existing off-the-shelf Embodied CLIP models [25] which have recently achieved SoTA performance on both of these tasks. We demonstrate significant improvements in these tasks using limited help from the expert. In ObjectNav, we improve 52% to 86% using 13% help, and in Rearrangement we go from 7% to 90.4% with 39% expert help. Importantly our ASK4HELP policy is model agnostic and can be easily applied to other E-AI models. Just as it has provided improvements over current state-of-the-art, it should also provide enhancements to the best models in the future. We also present strong results when

using human experts (measured via human trials) and noise-corrupted versions of the algorithmic expert and show that our policy is robust to these variations.

## 2 Related Work

**Embodied AI.** Recently, various embodied tasks such as navigation [4, 24, 21, 19, 31], instruction following [45, 2, 29, 16], manipulation [15, 55, 58], embodied question answering [13, 20], and rearrangement [52, 3] have witnessed tremendous progress. This can be attributed to the availability of open-source benchmarks [45, 2, 46, 51, 40, 22] and simulators [27, 32, 38, 44, 18, 55], stronger visual backbone models [25], self-supervised auxiliary tasks [57, 56] and task-specific inductive biases such as semantic mapping [7, 6]. Although, [54] show that the task of Point Navigation [2] in unseen environments can be solved by training on 2 billion frames with reinforcement learning, the success rate of the best models for most embodied tasks is still very poor [25, 52, 41].

Complementary to the modeling progress, a line of work has emerged in the past few years that attempts to learn to request assistance in the form of language [35, 49, 37], sub-goals [34], and agent state and goals [33]. However, [34, 35] rely on imitation to learn when the agent should request help. This can be limiting since the human-defined criteria to label the help-requesting behavior might not be optimal from the agent's standpoint. [49, 37] collect human-human dialog for task completion to learn communication between the agent and the expert. We try to make requesting expert assistance a part of the reinforcement learning problem, and let our ASK4HELP policy infer the trade-off between the amount of assistance and performance. Along similar lines, [33] consider learning an "intention" policy to request varying modalities of information, such as object detections, room types, and sub-goals, from the expert. These requested modalities may not, however, be easy or well-suited for an actual human user to provide.

With similar motivation to our work, [10] propose two ways of asking for expert help, a heuristic criteria based on model confidence and an augmentation of the agent's action space. We implement the heuristic criteria and present a comparison using the RoboTHOR [14] object navigation (ObjectNav) task. They also propose augmenting the agent's action space with an extra *ask* action. However, this involves retraining the underlying embodied model and additional reward shaping to prevent degenerate solutions. In contrast, we just train an extra policy without modifying the underlying Embodied AI model. We provide assistance in the form of expert actions similar to [10].

**Active Learning and Perception.** Our method for learning when to provide assistance based on the agent's internal representation is analogous to active learning [43]. In active learning, the agent tries to acquire labeled data to improve its model in the most sample-efficient manner [17, 42]. However, contrary to the general active learning paradigm, we do not collect additional data to train our agent. Instead, we try to learn a policy with minimal information about the underlying Embodied AI model to answer the question, within an episode, when is the expert intervention most useful? We draw inspiration from active learning in the sense that the model tries to pick out points in time for expert intervention as efficiently as possible. Prior works [47, 26, 48] use pre-defined metrics and heuristics to define state discrepancy and ask for demonstrations. However, these heuristics fail to generalize in unseen environments. We show the efficacy of our method in unseen, visually rich 3D environments [14, 52].

Uncertainty estimation using the value function to provide help and improve the sample-efficiency of RL algorithms has been explored in the literature before [12]. [50] propose a teacher-student framework to study how an RL agent can transfer knowledge to another agent by action-advising. [11] extend this idea to a multi-agent setting where agents learn to teach and advise simultaneously. These works lay down some foundational ideas to involve expert in-the-loop, but they mostly focus on improving the sample efficiency and knowledge transfer of RL algorithms. Contrarily, we propose to learn how to utilize human intervention to improve an off-the-shelf embodied agent's performance.

Recently, active perception has also been studied in the Embodied AI community. [8] learn a policy to choose which frames to label for object detection. [5] use perception models learned from internet data to learn an active exploration policy. [28] propose to fine-tune the object detection model at test time while interacting with an environment. [36] learn an exploration policy that balances the trade-off between semantic segmentation performance and the amount of annotation data requested. In contrast, we do not assume access to the underlying perception or planning parameters of the agent, we try to learn when human intervention would be effective in completing the task.

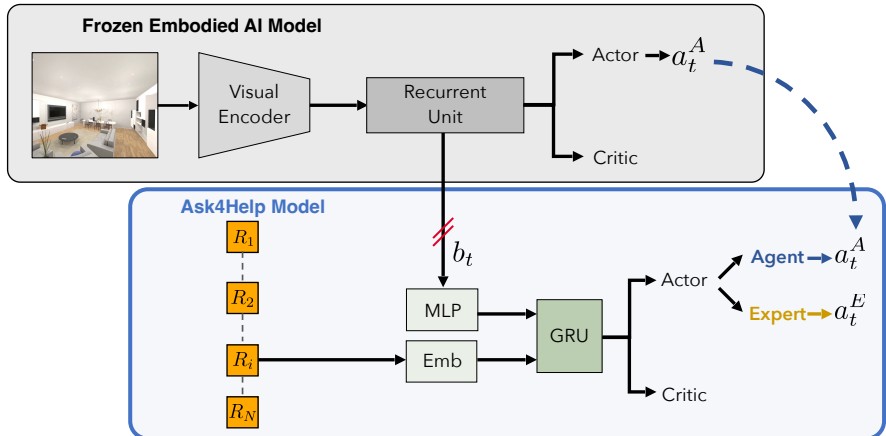

Figure 2: **Model overview.** The ASK4HELP policy selects if the next action should be taken by the E-AI agent or the expert. $b_t$ represents agents "beliefs", *i.e.* the recurrent unit's hidden state. $a_t^A$ denotes the action at time $t$, the superscript $A$ indicates it was produced by the underlying agent's policy. $E$ denotes the expert. The double red lines indicate that gradients do not flow through that part of the model. *Emb* represents a simple look-up embedding layer. $R_{1:N}$ represent different reward configurations that we embed (see Sec. 3.3).

## 3   Learning to Leverage Expert's Help

Today's state-of-the-art E-AI models are not yet performant and reliant enough to be deployed into real applications. We propose ASK4HELP, a policy to augment E-AI models with the ability to ping an expert, such as a human being, and leverage their intermittent help for the desired task.

### 3.1   Problem Definition

Consider a typical Embodied AI (E-AI) model as shown in Figure 2. It consists of a visual encoder, a state encoder (typically a recurrent unit), and an actor-critic head, trained using Reinforcement Learning (RL). We wish to augment this model with the ability to query an expert for help. Asking for too much help can be costly, or annoying if the expert is a human being. Hence, our goal is to achieve a good trade-off between asking for a minimal amount of help and maximizing the performance of the agent at its task. We propose a framework to learn a policy that decides when it is most effective to ask for help. We refer to this policy as the ASK4HELP policy. Importantly, we learn this ASK4HELP policy, without modifying the weights of the underlying E-AI model, allowing us to easily augment any off-the-shelf model with this capability. For this work, we consider that the help is provided in terms of the optimal action at the next time step, denoted by $a_t^E$ in Figure 2.

### 3.2   ASK4HELP Policy

The output of the ASK4HELP policy determines if the agent or the expert policy would act at a particular time step $t$. The objective of this policy is to maximize the rate of successful task completion while minimizing the amount of expert intervention. The ASK4HELP policy outputs, at every step, one of two actions (*Agent* or *Expert*) indicating whether the agent or expert should take control at that time step. Hence, when the ASK4HELP policy outputs the *Expert* action, the action on the next time step is taken by the expert i.e. $a_t^E$.

ASK4HELP model's architecture is depicted in Figure 2. It consists of a multi-layer perceptron and a GRU cell. It takes the underlying E-AI model's GRU hidden state (hereafter referred to as **beliefs**) $b_t$ as an input. The ASK4HELP policy is trained using reinforcement learning, specifically DD-PPO [54]. It receives two negative penalties, a relatively large one for task failure, and a smaller penalty for requesting expert help. The RL loss tries to balance the trade-off between these two penalties, thereby attempting to avoid failure with minimal expert help. The gradients from this loss are used only to update the ASK4HELP policy, recall that the agent's policy is frozen. We describe the reward structure and training details in Section 4.

## 3.3 Adapting to User Preferences at Inference Time

The methodology described in Section 3.2 is an effective way to achieve a good trade-off between task success and the amount of help requested. The amount of help requested by the agent is governed by the reward structure used during RL. Requiring the agent to ask for help less (or more) frequently would require us to retrain the ASK4HELP policy. This need for retraining is limiting as we may wish to deploy the same system in multiple settings each with very divergent costs associated with querying the expert: for instance, some users might be happy to provide feedback during the day while others might find such requests highly disruptive. Ideally, such users should be able to communicate their preferences to the agent and have it query for help at a rate that respects those preferences.

A naïve way of tackling this would be to train multiple ASK4HELP policies with a range of potential user preferences and deploy all of them onto the robot. The user could then choose the agent most aligned with their preferences. Unfortunately, the cost of this approach grows linearly in the number of user preferences we wish to capture, fails to share information across ASK4HELP policies, and can only ever represent a discrete collection of preferences.

Therefore, to support a wider range of user preferences while decreasing computational costs, we propose to explicitly condition our ASK4HELP policy on user preferences during training. In particular, as shown in Figure 2, we sample a range of reward configurations $R_{1:N}$ representing potential user preferences with different penalties associated with agent failure. Then, we train our agent by sampling rollouts with each reward configuration $R_i$ with uniform probability. We sample $R_i$ for a particular episode and keep it fixed throughout that episode. As shown in Figure 2, we embed the randomly sampled reward configuration $R_i$ into a 12-dimensional vector and provide that as an additional input to the ASK4HELP policy's GRU. This allows the ASK4HELP policy to modify its behavior according to user preference. We show in Section 4 that this method allows us to train a single ASK4HELP policy that can adjust the amount of help based on the reward configuration embedding.

# 4 Experiments and Analysis

We present empirical results and analyses when using our ASK4HELP policy with agents trained to complete the RoboTHOR Object Navigation (ObjectNav) and Visual Room Rearrangement (RoomR) tasks. We describe our experimental setup and the baselines in Sections 4.1 and 4.2, respectively, and present quantitative evaluations of ASK4HELP and competing baselines in Sections 4.3 and 4.4.

## 4.1 Experiment Setup

In our experiments, we wish to capture the setting in which a practitioner has access to a frozen off-the-shelf Embodied AI model for a particular task and wants to extend this model by enabling it to ask for help. To be able to add this ask-for-help capability, we assume the practitioner has a newly created (or held-out) set of training data that was not used to train the off-the-shelf model and that this new training dataset is possibly significantly smaller than the off-the-shelf model's training set. To this end, in what follows we will retrain existing SoTA models for ObjectNav and RoomR on subsets of their original training datasets, freeze these models, and then use the held-out training data to train a lightweight ASK4HELP policies that works in conjunction with the E-AI models.

**Dataset Split.** We train the RoboTHOR [14] ObjectNav and iTHOR 1-phase RoomR [52] models proposed in EmbCLIP [25], currently the published SoTA models for these two tasks, on 75% of the training scenes for their respective tasks (45 scenes for ObjectNav and 60 for RoomR). We use the publicly available codebase[5] provided by [25]. With these models fixed and frozen, we use the remaining 25% training scenes to train the the ASK4HELP policy. We evaluate our models on the unseen validation scenes that the agent has not seen before in training.

## 4.2 Baseline Definitions

We present a comparison of our ASK4HELP framework with various baselines, which are defined as:

- *Naive Helper (NH)* : The agent receives expert help at a step with probability $p$. In particular, at

---

[5]https://github.com/allenai/embodied-clip

every step, the NH samples from a Bernoulli($p$) distribution and takes the agent's action if it samples a 0 and, otherwise, takes the expert action. We control the amount of help by varying $p$ and produce different variants of this baseline. We denote each variant using NH-$p$.

• *Model Confusion (MC)* : This baseline is based on a heuristic criteria proposed in [10] for requesting expert assistance. Intuitively, MC will request help when the agent is not sufficiently confident that there is a single correct action to take. More specifically, MC will mark an agent as confused (thus requiring expert assistance) at a time step $t$ when

$$p_{sorted}^t[0] - p_{sorted}^t[1] < \epsilon, \tag{1}$$

where $p_{sorted}^t$ are the action probabilities at time $t$ sorted in descending order. We vary the $\epsilon$ parameter to produce different variants of this baseline and refer to them as MC-$\epsilon$.

These baselines do not assume access to the underlying E-AI model parameters. As mentioned in Section 3.1, our ASK4HELP policy follows the same constraint. However, note that for the *Naive Helper* and *Model Confusion* the underlying E-AI model is the SoTA model from [25] trained on 100% of the training scenes. Whereas, for our ASK4HELP policy, as mentioned previously, we use the same model architecture trained on 75% training scenes as the underlying E-AI agent. We use the remaining 25% training data for the ASK4HELP policy.

### 4.3 RoboTHOR Object Navigation

#### 4.3.1 Task Description and Metrics

**Task.** ObjectNav requires an agent to navigate through an environment and find an object of the specified category. The open-source RoboTHOR [14] environment supports this task with a robotic agent placed into a visually rich home environment. The agent starts at a random location and is given a target object category (e.g., *apple*) to find. Its action space consists of `MoveAhead`, `RotateRight`, `RotateLeft`, `LookUp`, `LookDown` and `End`. An episode is considered successful if an instance of the target object category is visible and within 1m of the agent.

**Metrics.** We report the success rate (SR) and SPL [1] for our navigation agents. We also quantify the amount of help with an Expert Proportion (EP) metric, which is defined as

$$EP = N_{expert}/N_{total}, \tag{2}$$

where $N_{expert}$ is the number of expert steps, and $N_{total}$ is the total number of steps in an episode.

#### 4.3.2 Training Details

**ASK4HELP policy.** We train the ASK4HELP policy using DD-PPO [54] for 15 million steps. We use the NavMesh algorithm in RoboTHOR [14] environment as the expert to request help from. It provides the optimal action from the shortest path to the agent's current location to the target. In practice, one does not have access to such an expert during testing/inference. Hence, we not only provide test results with this NavMesh expert but also provide results with a noisy version of the NavMesh expert as well as a human expert.

We use the AllenAct [53] framework to implement our model and training pipeline. The reward at any time step $t$ is defined as:

$$r_t = r_{fail} + \mathbb{1}_{ask} \cdot r_{init\_ask} + r_{step\_ask},$$

where $r_{fail}$ is a negative penalty for failure, $r_{init\_ask}$ is a one-time penalty given to the agent when it requests expert help for the first time with $\mathbb{1}_{ask}$ the indicator function controlling this one-time negative reward, and $r_{step\_ask}$ is a smaller penalty given for every step the expert takes. For ObjectNav, we set $r_{fail} = -10$, $r_{init\_ask} = -1$, and $r_{step\_ask} = -0.01$.

The $r_{init\_ask}$ penalty encourages autonomous operation when possible by discouraging the ASK4HELP policy from requesting help during an episode where the agent may be able to complete the entire episode without any assistance. Additionally, once expert help has been requested during an episode, $r_{step\_ask}$ encourages the policy to minimize the number of requests made to the expert. Here $r_{step\_ask}$ which is made significantly smaller in magnitude than $r_{init\_ask}$ to reflect the intuitive assumption that the marginal cost of assistance decreases with repeated queries (e.g., if a human has been contacted for assistance, then the cost of giving two expert actions is very similar to just giving one).

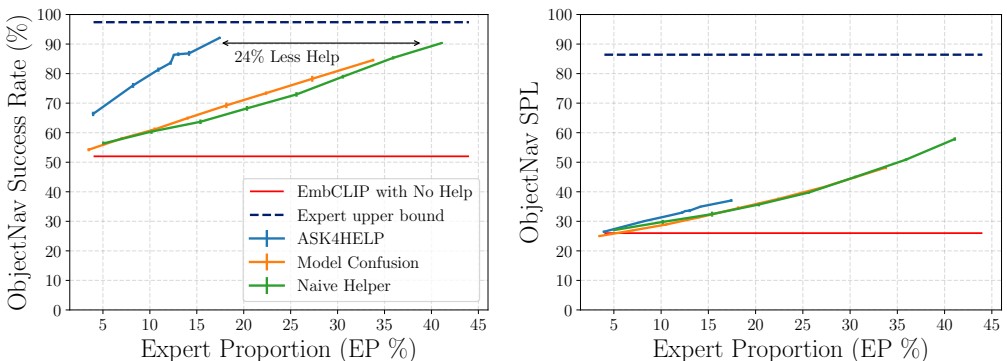

Figure 3: **ASK4HELP policies utilize expert queries efficiently for ObjectNav.** By varying the $\epsilon$ parameter of Model Confusion and Naive Helper baselines as well as the reward embedding input to the ASK4HELP policy (recall Sec. 3.3), we obtain a collection of different trade-offs between the proportion of steps spent querying the expert (EP) and the performance of these models as measured by the success rate and SPL metrics. As these plots show, our proposed ASK4HELP policies dominate the other methods, achieving better success and SPL values for all EP values. These results shown here are an average of 5 evaluations of each model on the ObjectNav validation set; while very small, we also show error bars corresponding to 1 standard deviation computed across the 5 evaluations.

It is also worth noting that we can vary $r_{fail}$ to generate multiple reward configurations $R_{1:N}$ to train a single policy as described in Section 3.3. Specifically, if we set $r_{fail} = -1$, which would imply that the cost of failure is not very high for the user, we get a reward configuration $R_{-1}$ with $r_{fail} = -1$, $r_{init\_ask} = -1$, and $r_{step\_ask} = -0.01$. Following the same trend, we vary, $r_{fail}$ from -1 to -30 to cover a broad range of user preferences and generate rewards configuration namely $R_{-1}$ to $R_{-30}$.

As discussed in Section 3.3, to train an agent with all reward configurations $\{R_{-1}, \ldots, R_{-30}\}$, we uniformly sample a reward configuration $R_i$ for each episode in the environment. The reward the agent receives during that episode is based on $R_i$.

During validation, we do not have access to these rewards, therefore we need to learn a correspondence between user preference and agent behavior. To accomplish that, we associate each reward configuration with an index (e.g. $-1 \rightarrow R_{-1} \rightarrow r_{fail} = -1$), and embed these configuration indices using a standard lookup in a learnable embedding matrix. We provide this embedding as an input to the agent. This allows the agent to learn a correspondence between this embedding input and the cost of failure.

### 4.3.3 Quantitative Analysis

**ASK4HELP results.** We present a comparison of our ASK4HELP framework with the baselines defined in Section 4.2. For these results, we train our ASK4HELP policy just once with multiple reward configurations as described in Section 3.3. This allows us to produce different variants of our method by simply varying the reward embedding input during inference. Figure 3 shows the performance (as measured by the success rate and SPL metrics) of our model against competing baselines when restricted to various EP values. As shown in the figure, our proposed ASK4HELP policy efficiently uses a limited number of expert queries to achieve high success (4% EP results in a 16% absolute gain in success and with just 17% EP the agent achieves >92% success) and strictly dominates the other baselines by obtaining higher performance at every EP level. In fact, to achieve greater than 90% success with the Naive Helper strategy, one must increase the EP to 41%, more than double what is required when using our ASK4HELP policy. Interestingly, the gains in SPL are somewhat less dramatic than for success rate. This is, however, intuitive: a high-quality ASK4HELP policy should only query the expert if it has already searched the scene exhaustively (thus attaining low SPL) and is certain it cannot find the object without assistance. Note that, due to environment noise the NavMesh [23] expert, denoted by the dark blue dashed line, does not achieve a 100% success rate. We provide qualitative results in the supplementary materials.

**Algorithmic *vs*. Human Expert.** In the above evaluations and when training our ASK4HELP policy we have used a NavMesh [23] expert which computes expert actions using ground-truth shortest paths from the environment. While such an expert may be available during training, no such expert is likely to be available at inference time as, otherwise, we may as well use such an agent rather than a learned model. This raises an important question: if the expert available at inference time may be different than the expert used at training, is our ASK4HELP policy robust to changes in the expert? To answer this question we conducted a human trial.

In this trial we selected a set of 131 episodes from the RoboTHOR ObjectNav validation set and, for each of these episodes, evaluated our ASK4HELP policy when using as an expert: (a) a set of 11 humans (each human expert was asked to provide assistance for between 10-15 episodes, details in supplement), (b) the standard NavMesh agent described above, and (c) a collection of "corrupted experts" CE-$\epsilon$ where, on being queried, a CE-$\epsilon$ agent returns either the correct action with probability (1-$\epsilon$) or a randomly selected navigation action with probability $\epsilon$. The results of our ASK4HELP policy in these trials are summarized in Table 1. Note that the performance of the ASK4HELP model when using the human and NavMesh experts is very similar, for

Table 1: **Human Trial Results**. SR, SPL and EL represent the Success Rate, Success by path weighted length and episode length metrics respectively. Expert column indicates the expert used. EP corresponds to the Expert Proportion metric.

| Expert | EP (%) | SR (%) | SPL (%) | EL |
|---|---|---|---|---|
| Human | 12.27 | 81.6 | 24.3 | 211 |
| NavMesh [23] | 11.09 | 87.7 | 21.9 | 239 |
| CE-0.1 | 10.18 | 82.4 | 22.7 | 235 |
| CE-0.2 | 11.77 | 84.0 | 23.0 | 234 |
| CE-0.4 | 12.01 | 80.2 | 21.5 | 252 |
| CE-0.8 | 19.88 | 61.8 | 18.2 | 293 |
| None | 0 | 55.7 | 22.3 | 184 |

approximately the same EP (12.27 *v.s.* 11.09) the humans appear to result in lower overall success rates (81.6% *v.s.* 87.7) but appear to guide the agent to using more efficient paths (SPL of 24.3 *v.s.* 21.9). Upon examining the failure cases of our model when using human experts we found many cases where humans would incorrectly end episodes when in sight of the target object but too far from the object to satisfy the RoboTHOR requirement that agents end their episodes within 1m of the target. The results with the CE-$\epsilon$ experts are somewhat surprising: even with significant expert corruption our ASK4HELP policy can still make use of the expert's feedback and obtains high performance with relatively small EP values. Even when $\epsilon = 0.8$, so that 80% of the expert's actions are randomly chosen, the ASK4HELP policy trained without any such noise is able to obtain success rates above the baseline model without help. We present CE-$\epsilon$ expert results on the full validation set in the supplement.

**Data efficiency of ASK4HELP.** As discussed in Section 4.1, we train the ASK4HELP policy on 25% of the training scenes. However, in some situations where training data is scarce, sacrificing this fraction might be unfeasible. To ablate the data efficiency of ASK4HELP policy, can we train ASK4HELP with 25% and 10% training scenes, with the exact same reward configuration and same underlying pre-trained task agent. We observe only a 2% drop in success rate and 1% in SPL for the same expert proportion (EP%). As the results indicate, ASK4HELP converges to a reasonable expert help-performance trade-off despite using just 10% training scenes, which in case of [14] object navigation is just 6 scenes.

**Replacing pre-trained agent with a random agent.** Having a pre-trained agent is important to ensure that we're not overusing the expert, and the underlying E-AI model is doing a major portion of the task. We train an ASK4HELP policy with a random object navigation policy, and compare it with an ASK4HELP policy that works with a pre-trained agent. Both the policies are trained with the same reward configuration and training regime. As shown in Table 2, a random agent takes far more expert help than a pre-trained agent, since it lacks to ability to perform object navigation hence is completely reliant on the expert.

| Model setting | SR | SPL | EP (%) | EA |
|---|---|---|---|---|
| ASK4HELP (pre-trained) | 86.3 | 33.2 | 12.31 | 17 |
| ASK4HELP (random) | 76.44 | 67.65 | 98.99 | 27 |

Table 2: Random vs Pre-trained task agent. SR denotes success rate, SPL denotes Success by path weighted length. EP denotes the expert proportion metric. EA denotes the number of actions taken by the expert.

**Comparing ASK4HELP and 'Model Confusion' baselines under constrained amount of help.** We performed an experiment where we make the expert available only for 20 steps in an episode

(both during training and evaluation) and train an Ask4Help policy and present a comparison with the model confusion baseline. On the unseen validation scenes, Ask4Help achieves a success rate of 70%, whereas the model confusion baseline (Section 4.2) achieves 61% success on object navigation.

## 4.4 Room Rearrangement in iTHOR

### 4.4.1 Task Description and Metrics

**Task.** We show results on the recently proposed 1-phase Visual Room Rearrangement (RoomR) [52] task. In RoomR, an agent is placed into a household environment, and given two egocentric images at every time step. One image shows the environment's current state, and the other image shows the target state that the agent must rearrange it to. The agent must then navigate around the room and interact with the objects to restore them to the goal state. The agent can take navigation actions `MoveAhead`, `RotateRight` etc. and high level interaction actions like `PickUpX`, where X is object category to be interacted with.

**Metrics.** We report the Fixed Strict (FS) and Success Rate (SR) metrics on the room rearrangement task. Please refer to the original work [52] for more details on these metrics. We also report the Expert Proportion (EP) metric described in Section 4.3.1.

### 4.4.2 Training Details

The ASK4HELP policy is trained using DD-PPO [54] for 15 million steps. The reward at any time step $t$ is defined as:

$$r_t = \mathbb{1}_{ask} \cdot r_{init\_ask} + r_{step\_ask} + r_{rearrange} \cdot$$

The $r_{init\_ask}$ and $r_{step\_ask}$ rewards serve the same purpose as they did in Object Navigation results presented before. We set $r_{init\_ask} = -0.5$, and $r_{step\_ask} = -0.02$. Unlike as for ObjectNav which used a sparse $r_{fail}$ penalty, we found that this sparse reward was insufficient to train our ASK4HELP policy (note that SoTA models fail at RoomR in ≈93% of cases). To this end, we replace the $r_{fail}$ penalty with the reward $r_{rearrange}$ which simply equals the rearrangement specific reward employed in [52]. We use the Heuristic Expert from [52] as the expert to request help from.

### 4.4.3 Rearrangement Results

We present the performance for our ASK4HELP policy and other baselines in Table 3. We show that using our ASK4HELP Policy, we can boost the performance dramatically in the Fixed Strict (from 18.13% to 95.32%) and Success Rate (from 6.9% to 90.4%) metrics, with an Expert Proportion of 39%.

Note that the Expert Proportion (EP) metric for rearrangement is higher than what we observe for Object Navigation. This can be attributed to both the task complexity, and the relatively poor performance of the SoTA EmbCLIP [25] model. However, since our ASK4HELP policy operates independent of the underlying E-AI model's parameters, our method would still be compatible with future models. If more performant models are proposed for the task, the ASK4HELP policy will accordingly adjust the amount of expert help requested to support that model effectively.

Table 3: **Room Rearrangement Results.** FS and SR represent the Fixed Strict (%) and Success Rate (%) metrics for rearrangement respectively. MC-$\epsilon$ and NH-p denote the Model Confusion and Naive Helper baseline variants.

| Model | EP (↓) | FS (↑) | SR (↑) |
|---|---|---|---|
| **ASK4HELP** (ours) | **39** | **95.32** | **90.4** |
| MC-0.2 [10] | 48 | 88.96 | 81.4 |
| MC-0.3 [10] | 63 | 94.64 | 89.9 |
| NH-0.4 | 40 | 85.97 | 77.5 |
| NH-0.5 | 50 | 90.24 | 83.6 |
| NH-0.6 | 60 | 92.77 | 87.6 |
| NH-0.7 | 70 | 94.85 | 89.6 |
| EmbCLIP [25] | 0 | 18.13 | 6.90 |
| **Expert** | **100** | **96.53** | **92.8** |

We also present a comparison against the *Naive Helper (NH)* and *Model Confusion (MC)* baselines described in Section 4.1. As shown in Table 3, we outperform all variants of these baselines with the same or higher Expert Proportion than our method. This highlights the ability of our method to effectively utilize minimal help to boost task performance. Notably, it takes a *Naive Helper* variant 70% Expert Proportion to perform on-par with our ASK4HELP policy that requires only 39% EP. Similarly, the Model Confusion baseline requires 63% EP to do the same.

# 5 Conclusion

We propose ASK4HELP policies, a model-agnostic approach to augment a given, pre-trained, E-AI model with the ability to query an expert for help. The goal of ASK4HELP policies is to leverage expert assistance to dramatically improve the reliability of existing embodied AI agents while also minimizing the number of requests made to these experts. Our approach is agnostic about the parameters of the underlying Embodied AI and shows that a relatively small amount of help results in massive performance improvements in object navigation and room rearrangement, two popular tasks in Embodied AI. By evaluating our ASK4HELP policies using different types of experts (shortest path planner, a noisy variation of the planner, and humans), we find that our trained ASK4HELP policies are robust to expert variation, desirable as it reduces the burden of having to use precisely the same expert during training and inference. In summary, ASK4HELP policies provide a generic and robust mechanism for improving the performance of E-AI models with minimal expert assistance.

**Limitations.** We highlight two limitations of our approach. First, while it is a great advantage that our ASK4HELP policies are model agnostic and do not require retraining the underlying agent, this has the disadvantage that our agent cannot learn from the expert's feedback. This means that the agent may query the expert multiple times from the same state, a frustrating user experience. Second, we have trained our policies using non-human experts available on-policy. While it is convenient that these experts exist for ObjectNav and RoomR, some E-AI tasks may only have offline datasets of expert trajectories collected from humans. Extending to this setting is exciting future work.

**Acknowledgements** We would like to thank members of AI2 PRIOR team for participating in the human trials. We would also like to thank Eric Kolve, Jiasen Lu and Kuo-Hao Zeng for helpful discussions and feedback. We would also like to thank the anonymous reviewers for their feedback and discussions. JC was partly supported by the NRF grant (No.2022R1A2C4002300), IITP grants (No.2020-0-01361-003, AI Graduate School Program (Yonsei University) 5%, No.2021-0-02068, AI Innovation Hub 5%, 2022-0-00113, 5%, 2022-0-00959, 5%, 2022-0-00871, 5%, 2022-0-00951, 5%) funded by the Korea government (MSIT).

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
