# OpenReview forum: "Ask4Help: Learning to Leverage an Expert for Embodied Tasks"
_NeurIPS.cc/2022/Conference — NeurIPS 2022 Accept_

### Official Review · Reviewer_bRWC · 2022-06-27

**Rating:** 4
**Confidence:** 4
**Soundness:** 1 poor
**Presentation:** 3 good
**Contribution:** 1 poor

**Summary:**

The paper introduces Ask4Help, a method for augmenting an existing policy with the ability to fall back to an expert policy during an episode. This is achieved without retraining the existing pre-trained agent by introducing a meta-controller that will select whether to follow the agent or the expert at every timestep. The meta-controller does not receive raw observations, but the agent's belief state, a prediction of the agent's success rate, and an embedding informing it about the user's preference (i.e. how costly it is to ask for help). The proposed method is evaluated on two tasks, namely object navigation (RoboTHOR) and room rearrangement (iTHOR), where it greatly boosts the success rate of the pre-trained agent while comparing favorably to other baselines in the amount of expert usage.

**Questions:**

**Major**

I have described my main questions and concerns in the previous section, namely:
- Stats about expert usage
- Additional baselines
- Importance of the different design choices in the Ask4Help architecture
- Dataset split


**Minor**

- After following the expert, the agent could find itself in states where it is out of distribution. This is a drawback of freezing the pre-trained agent that is not discussed in the manuscript, and a discussion about this would benefit the paper.
- Section 3.3 describes how Ask4Help can be trained with multiple user preferences. However, Section 4 describes a single reward function. Could you please explain how this is done, and provide examples showing how this affects the success rate and expert usage?
- Given the connections with Hierarchical RL, I would strongly recommend extending the Related Work section to include an overview of the field.

**Limitations:**

Yes.

**Strengths And Weaknesses:**

**Strenghts**

- The problem of providing agents with the ability to ask for help is an important one.
- The paper is easy to follow.


**Weaknesses**

- The method itself is not novel, as it can be seen as a Hierarchical RL with two low-level policies: the pre-trained agent and the expert.
- While the quantitative results look strong, the videos in the supplementary material show that the discovered behavior is extremely simple. During a first phase the meta-controller selects the agent, which does not seem to know how to solve the task and simply roams around the room. Then, it selects the expert for a few timesteps -- which finds the object and solves the task. This results in high success rate (thanks to the hard-coded expert) but low expert usage (which is diluded due to the long initial phase where the agent is used). It would be helpful if authors could provide some statistics about expert usage aggregated over all tasks (e.g. detailed stats about the timesteps in which the expert is used within an episode, aggregated over all episodes).
- In light of the aforementioned qualitative results, I am not convinced that the baselines are adequate. There are some additional baselines that would help understanding whether the Ask4Help policy is learning something trivial or not:
  - Replacing the pre-trained agent with a random and/or no-op policy. This would provide insight about whether it is really learning to combine the pre-trained agent and the expert, or whether the observed gains just come from the fact that the expert will always solve the task when given enough time.
  - Replacing the meta-controller with a hard-coded policy that selects the agent for M steps and then runs the expert for N steps (where both M and N should be swept over).
- The Ask4Help policy has a non-standard architecture, e.g. it takes the predicted success rate as input. There are no ablation studies in the paper (nor the supplementary material), which make it difficult to understand whether this is an important component of the agent or could be removed.
- The dataset split described in L208 is non-standard in machine learning. My understanding is that the purpose of the split is to deploy the agent on a set of tasks it has never seen before and where it might not perform well. However, why isn't there a third, truly held-out, set ot tasks where one can evaluate whether the Ask4Help policy generalizes? Otherwise, if one is allowed to train and evaluate on the same set of tasks, why can't we re-train the agent on the validation set instead of using Ask4Help?

---

> ### Author Response · Authors · 2022-08-02
> **Author Response 1 to Reviewer bRWC's comments**
>
> -   **Method as Instantiation of hierarchical RL**
>
> The analogy of Ask4Help being the high-level policy, and the underlying task agent and expert being the low-level policies can be seen as Hierarchical RL. However, in our case, neither the expert nor the underlying task agent is trained. We’ll clarify this distinction in the related works updated draft.
>
> -   **Clarification on Dataset Split.**
>
> Yes, there is a held-out **unseen validation** split that both the pre-trained agent and the Ask4Help policy have not seen or been trained on. All the results presented in the draft are based on this unseen validation set.
>
> Your understanding, that the purpose of the split is to deploy the agent on a set of tasks it has never seen before and where it might not perform well, is correct.
>
> We do precisely this: we train our Ask4Help policy on the 25% data from the **training** scenes and perform evaluation on the held-out **unseen validation** scenes which neither the pre-trained agents nor the Ask4Help policy have seen before. The fact that the Ask4Help policy shows good results in those **unseen validation** scenes is indicative of its generalization.
>
> -   **Expert usage statistics, pretrained agent’s role.**
>
> The underlying pre-trained agent is an off-the-shelf state-of-the-art ObjectNav agent which achieves 50% success in validation unseen scenes on RoboTHOR. We find that from 1800 episodes in unseen validation scenes, the Ask4Help policy does not provide expert intervention in 619. Out of these 619, 486 end up being successful (78.5%). The pretrained agent is performing a significant portion of the episodes autonomously. We show a [histogram here](https://anonymous-neurips22.s3.us-west-2.amazonaws.com/a4h/agent_without_expert.pdf) of the number of steps the agent takes in these 619 trajectories where Ask4Help does not invoke the expert.
>
> Additionally for cases where the Ask4Help policy does request expert help, we did some investigation to show that the underlying pre-trained agent is doing non-trivial exploratory work before the Ask4Help policy requests expert help. We plot the closest distance to the target object for the pre-trained agent before it asks for help against the distance of the object from the agent's initial position. We plot the same curve for a random agent for the same episodes on the unseen validation set. The curve can be found here: [plot here](https://anonymous-neurips22.s3.us-west-2.amazonaws.com/a4h/min_distance_to_tgt_curve.pdf). As the curve indicates, the pre-trained agent gets closer to the target object (before asking for help) more frequently than a random agent.
>
> -   **Stats about time steps in which expert is used**
>
> We present a plot showing the number of steps before the expert is invoked for the first time by our Ask4Help policy with a pre-trained agent and a random agent. The plot is [here](https://anonymous-neurips22.s3.us-west-2.amazonaws.com/a4h/expert_timestamp_plot_absolute.pdf). The Ask4Help module with the pretrained agent frequently waits many time steps before invoking the expert whereas Ask4Help with the random agent will almost always request expert help within the first 5 steps.
>
>
> -   **Discussion on emergent behavior**
>
> The emergent behavior is quite diverse. For instance, in one example, the agent has to go around a chair to get a good view of the laptop, and the Ask4help policy intervenes to get the agent around that situation. In another case, it fails to find the Alarm Clock because it is on top of the cabinet and hence out of view. The expert help allows the agent to look up and successfully finish the task. Note that, as discussed in Section 4.3.2, the $r_{init\\\_ask}$ penalty discourages the Ask4Help policy from requesting help at all during the episode and allows autonomous operation. Therefore, this emergent behavior is not surprising as the Ask4Help policy allows the agent to attempt the task for a reasonable period before requesting expert help, which in some cases can be the entire episode itself. As suggested, we also present a plot showing at which time step the expert takes over for the first time [here](https://anonymous-neurips22.s3.us-west-2.amazonaws.com/a4h/expert_timestamp_curve_absolute.pdf).

---

> ### Author Response · Authors · 2022-08-02
> **Author Response 2 to Reviewer bRWC's comments**
>
> -   **There are some additional baselines that would help understanding whether the Ask4Help policy is learning something trivial or not**
>
>
> We discuss the suggested baselines and our findings below.
>
>
> Baseline 1: Replacing pre-trained agent with Random/No-op agent - We implement this baseline and train the Ask4Help module with a random underlying agent. Note that the Ask4Help attempts to maximize task performance with minimal expert help.
>
> The results on unseen validation scenes is as follows:
>
>
> | Model setting                | SR    | SPL   | EP (%) | Number of Expert Actions |
> |------------------------------|-------|-------|--------|--------------------------|
> | Ask4Help (pre-trained agent) | 86.3  | 33.2  | 12.31  | 17                       |
> | Ask4Help (random agent)      | 76.44 | 67.65 | 98.99  | 27                       |
>
>
>
> In the case of a random agent, since its object navigation performance is really low, the Ask4Help policy converges to asking for very high expert help (27 expert steps on average per episode) to prevent task failure and still achieves only 77% task success.
>
> With a pre-trained agent (in this case Khandelwal et al. EmbodiedCLIP[25]), Ask4Help can achieve 86% task success with just 12% expert help. The pre-trained model is playing a good part in achieving high performance complimentary to the expert help.
>
>
>
> Baseline 2: Replace meta-controller with hard-coded policy -
>
> We implement the suggested hard coded policy, we run the pre-trained agent for M steps and then the expert for N steps, after **M+N** steps, the episode ends. For both M and N, we try the following values, [10,20,30,40].
> **M - Number of agent steps, N - Number of Expert steps.**
>
>
>
> | M ↓  \ N → | 10   | 20    | 30    | 40    |
> |:----------:|:----:|:-----:|:-----:|:-----:|
> | **10**         | 23.8 | 46.2  | 66.2  | 82.3  |
> | **20**         | 28.2 | 48.67 | 68.72 | 72.72 |
> | **30**         | 30.7 | 49.3  | 68.5  | 81.4  |
> | **40**         | 33.7 | 50.1  | 68.2  | 80.9  |
>
> A policy trained with Ask4Help uses an average of 17 expert steps and 157 agent steps, and achieves a success rate of 86.3.
>
> -   **Success Prediction as an important component of the agent?**
>
> Based on experimentation, we find that success prediction provides a useful way of capturing failure modes of simpler tasks like object navigation and can provide a strong learning signal. However, for complex tasks like rearrangement, it is difficult to capture failure situations with a simple success classifier. This motivates us to give the agent belief as input, which is important to generalize to more complex tasks.
>
>
>
> -   **After following the expert, the agent could find itself in states where it is out of distribution.**
>
>
> Yes, this is a possibility, and could be a potential drawback of freezing the pre-trained agent. However, since we are training the Ask4Help policy with RL to intervene with expert actions, it could potentially learn to provide help in situations that would not push the agent into unknown states.
>
> This is an interesting consideration for future work, and we will modify the draft to include discussion on this.
>
>
>
> -   **Section 3.3 describes how Ask4Help can be trained with multiple user preferences…**
>
>
> We clarify this in the [common response here](https://openreview.net/forum?id=_bqtjfpj8h&noteId=_Ro-f-VRKCe).

---

> > ### Comment · Reviewer_bRWC · 2022-08-08
> > **Discussion**
> >
> > Thanks for the detailed response, it addressed some of my major concerns (e.g. about the data split). I will update my score accordingly, but I would like to ask some follow-up questions first.
> >
> > **Baseline 1: Ask4Help with random agent**
> >
> > What are the task and the success rate of the pre-trained agent? I find the EP metric here to be somewhat misleading: an increase of 10 expert steps (from 17 to 27, a 58% relative increase) makes this metric go from 12% to 99%. Is this because Ask4Help with a pre-trained policy learns to wait before asking for help due to the reward structure? Why doesn't this happen with the random policy? The SPL metric is doubled when replacing the pre-trained agent with a random one; if I understood this correctly, this means that the agent does not wander around so much.
> >
> >
> > **Baseline 2: hard-coded meta-controller**
> >
> > I am surprised by the low scores here, as well as by the fact that the success rate is mostly determined by the value of N. [This plot](https://anonymous-neurips22.s3.us-west-2.amazonaws.com/a4h/min_distance_to_tgt_curve.pdf) shows that the initial distance to the target is 10, why is the success rate so low for values of N such as 20 or 30? Finally, the values chosen for M and N seem to differ substantially from what the Ask4Help policy generally does (with M=157 and N=17). I wonder if authors have experimented with values for M and N in that ballpark.
> >
> >
> > **Impact of success preditiction**
> >
> > If I understood replies to other reviews correctly, it looks like success prediction does not play a very important role in the final performance of the agent (i.e. the belief state is enough). Since success prediction is presented as an important part of the proposed method, but seems to have little to no effect on the results, the paper should be re-written and updated to reflect this finding. I am concerned about the magnitude of the change required, which might be too big for a rebuttal period.

---

> > > ### Author Response · Authors · 2022-08-09
> > > **Author Response to Discussion comments and questions**
> > >
> > > We thank the reviewer for their time and response to our rebuttal. The suggested experiments were very insightful and have certainly improved our work. We address the follow up questions below.
> > >
> > > - **What are the task and the success rate of the pre-trained agent?**
> > >
> > > For object navigation, the pre-training task is object navigation, and the success rate is 49%. For rearrangement, its rearrangement and the success rate is 7%. We use the EmbodiedCLIP [25] models from Khandelwal et al. as the pre-trained policies.
> > >
> > > - **I find the EP metric here to be somewhat misleading: an increase of 10 expert steps (from 17 to 27, a 58% relative increase) makes this metric go from 12% to 99%. Is this because Ask4Help with a pre-trained policy learns to wait before asking for help due to the reward structure? Why doesn't this happen with the random policy?**
> > >
> > > Great question! This is due to the episode lengths being different for the agents. Since the random agent is very bad at doing the task, the expert takes over very early on, and tries to take the shortest path and hence the episode is terminated sooner as compared to a pre-trained agent. The pre-trained agent explores the scene so as to attempt to solve the task on its own.
> > >
> > > The Ask4Help reward structure encourages the agent to complete as much portion of the task autonomously as possible. This does not happen with random policy since it is very bad at performing the task, the only way to reach a favorable performance trade-off is for the expert to do most of the task. Whereas, the pre-trained agent has a success rate of 49% which makes it very good at completing a major portion of the task, so Ask4Help holds off expert help until it is absolutely needed.
> > >
> > > However, it is important to note that Ask4Help does not necessarily always wait for a long time, it can provide some help early in the episode.
> > >
> > > - **I am surprised by the low scores here, as well as by the fact that the success rate is mostly determined by the value of N.  [This plot](https://anonymous-neurips22.s3.us-west-2.amazonaws.com/a4h/min_distance_to_tgt_curve.pdf)  shows that the initial distance to the target is 10, why is the success rate so low for values of N such as 20 or 30?**
> > >
> > > Note that the initial distance to the target is not always 10 meters, as it can be seen in the curve, it can have values in the range of 0-10.
> > >
> > > The success rate is mostly governed by N since this heuristic does not allow the expert to intervene in agent exploration at all.
> > >
> > > For the low scores, we believe that the underlying pre-trained agent generally requires more time to explore and converge closer to the object so it might go to a far off location initially after 20-30 steps, recovering from that and finding the target might require more than 30 expert steps. It is also a possibility that the agent requires some help during this exploration if it gets stuck or is looping in one particular region. This is also an indication that it is ideal if the pre-trained is allowed to explore autonomously as much as possible, it can explore most of the space and usually requires help in getting around immovable obstacles or general recognition failures.
> > >
> > > - **Finally, the values chosen for M and N seem to differ substantially from what the Ask4Help policy generally does (with M=157 and N=17). I wonder if authors have experimented with values for M and N in that ballpark.**
> > >
> > > We tried an experiment with the value of M=160 and N=40, and that achieved a success rate of 72.8 (as compared to Ask4Help which gets 86.3 with M=157 and N=17). We believe this is an interesting insight as it tells us that naively allowing the agent to go for a certain amount of steps, and then just using the expert is also not as effective as our learned Ask4Help Policy that learns to combine the pre-trained agent and expert in a way that is very performant. We'll include this in the updated draft. We thank the reviewer for this comment.
> > >
> > > - **Impact of Success Prediction**
> > > The insight that belief is sufficient is a very useful one and we assure the reviewer that we will update our draft to reflect that accordingly. It is a benefit of using OpenReview and engaging in active discussions with reviewers.
> > >
> > > However, we would like to highlight that the core contribution of our work, learning an auxiliary-policy that can enable reusing off-the-shelf embodied AI models, is useful and important as you've pointed out.

---

### Official Review · Reviewer_rsmr · 2022-07-10

**Rating:** 4
**Confidence:** 4
**Ethics Flag:** Yes
**Soundness:** 2 fair
**Presentation:** 3 good
**Contribution:** 2 fair

**Summary:**

This paper introduces a method, Ask4Help, for incorporating expert knowledge for embodied AI. The authors leveraged a pre-trained embodied AI agent and designed a policy to switch between agents' prediction and expert action as the final acting policy. The switching policy is trained by RL given  the reward from both success rate and the portion of expert knowledge used in the episode. The resulting human-in-the-loop policy achieves performance improvement on several common embodied AI challenges.

**Questions:**

1. As mentioned previously in the Weakness section, the authors should clarify the motivation of designing a policy that directly adopts expert knowledge without learning from it, especially on how such a policy would benefit us on more embodied AI challenges without that much expert knowledge.
2. The authors should discuss the design of experimental settings and baseline methods: why not set a constraint of expert knowledge available? why not compare with prior methods that refine models with human expert knowledge under the same constraint? how is the current result significant since it is actually provided with all expert knowledge since the model learns to select between using it and not using it automatically?
3. Model-wise, the motivation for designing the Success Prediction Model (SPM) is somewhat duplicative to me since it is in-essence doing the same job of predicting which action to use (e.g. the expert's when success prediction is low, the model's when success prediction is high). I hope the authors could discuss if the SPM model is substitutable by the reward received by Ask4Help since it is trained offline with ground truth successful and failure trails.
4. In figure 3, why is the upper bound of results, not 100%, I guess this is related to the task but hope the authors would clarify.

**Limitations:**

The authors have partially stated their limitations on continual learning. However, I think there is still more work to do to make the current experimental design solid and sound. I hope the authors would consider the facts stated previously in Weakness and Questions, especially on the significance of the results since it now needs all expert knowledge during training and adopts partial of them during acting. This setting, to me, is not reasonable and might need better adjustment.

**Strengths And Weaknesses:**

[+] The problem of incorporating expert knowledge and collaborating with humans is becoming increasingly important over the past few years, especially with more agents showing strong potential for indoor tasks. This makes the main topic of this paper important and meaningful.

[+] The overall writing of this paper is clear and illustrative with ideas, methods, and results clearly stated. The authors showed significant performance improvements on several embodied AI challenges by augmenting a pre-trained embodied AI agent with expert knowledge. This shows strong potential for the pipeline of prompting large-scale pre-trained agents with expert knowledge, especially with recent trends on language understanding (e.g. few-shot capabilities of GPT-3).

[-] The major concern of this paper comes with its design and evaluation metrics. The authors stated in the limitation section that the proposed model possesses the disadvantage that it can not learn from the expert's feedback. This, in my opinion, is a critical issue since it always requires expert knowledge to perform well and fails to leverage extra in-context expert knowledge. As the current policy only switches between the pre-trained agent's action and the expert's action, I don't see a way to make the current system a self-adaptive one when put into a new training scene. This makes the whole point of designing such a policy for querying expert knowledge questionable.

[-] Following the previous point, I do think the current experimental settings and results are not convincing or have little impact. The authors used RL to train the Ask4Help policy for deciding which action to use (expert's or the model's) and seem to provide full accessibility of expert knowledge when generating the final scores (since the model can choose the amount of expert knowledge to use), this makes the current results hard to interpret as the model can always learn to use more expert knowledge without a limit. In this case, a proper comparison in my opinion should be constrained under the amount of expert knowledge available and over the performance gain of applying each method. Next, the baseline models are also designed to follow the same pipeline of policy switching without baselines from previous works that leverage the same amount of data for imitation learning or refinement.

---

> ### Author Response · Authors · 2022-08-02
> **Author Response to Reviewer rsmr's comments and questions**
>
> We thank the reviewer for insightful and detailed comments. We appreciate the positive comments on the importance of the problem, significant performance improvements, strong potential, and clarity of writing. We will now respond to the comments and questions raised in the review.
>
> -   **Clarification on motivation of problem setting**
>
> The ability to ask for help is important with or without test-time adaptation. Clear examples of this are any safety-critical tasks; for instance, currently deployed self-driving cars have proprietary heuristic mechanisms for detecting when the system is uncertain and hand control back to the expert human driver. Beyond these safety-critical settings, asking for help can be used to take a model that is not ready for real-world deployment, because its performance is too low, and immediately make it deployable by injecting expert help. While the area of test-time adaptation is very exciting, it is still a relatively immature technology.
>
> It is worth mentioning also that there are many potential deployed systems where test-time adaptation may be beyond the limitations of the available hardware (e.g. a cpu-bound house cleaning agent).
>
> Building self-adaptive agents is an interesting research question and our work is a step towards achieving that goal. It is part of the future work that we’re considering. However, having the ability to prompt off-the-shelf Embodied AI models in an efficient manner with expert knowledge is a useful tool given that test-time adaptation is still an active problem.
>
> -   **Comparison with a constrained amount of help.**
>
> We performed an experiment where we allow a maximum of only **20** expert steps in an episode (both during training and evaluation) and train an Ask4Help policy and present a comparison with the model confusion baseline. On the unseen validation scenes, Ask4Help achieves a success rate of **70%**, whereas the model confusion baseline we present in the draft (Section 4.2) achieves **61%** success on object navigation. We believe this is a useful insight and would happily include it in the updated draft. Thanks for the suggestion.
>
> -   **Success Prediction Ablation**
>
> In particular, when we mask out the belief input for Rearrangement, we find that the Ask4Help policy simply learns to request expert help 97% of the time, which is very undesirable from the user perspective. This is due to the complex nature of rearrangement which makes it much harder for success prediction to capture the modes of failure where the agent might require assistance. However, for simpler tasks such as object navigation, success prediction provides an effective learning signal, since it is able to capture failure situations better.
>
> We believe this indicates that having the agent belief as an input allows our method to be generally applicable to tasks of varying difficulty, especially for complex ones like Rearrangement where purely relying on success prediction may not be enough.
>
> -   **Upper bounds of result is not 100%**
>
> The expert available in RoboTHOR is not perfect and has some edge cases where it fails. Notably in cases where the object is on top of some cabinets or shelves and is difficult to bring in view. Hence, it does not achieve 100% success on validation scenes, but these cases are very rare.
>
> Similarly for rearrangement as shown in Table 2, we use a heuristic expert and the upper bound on expert performance is 92.8% since it also has some edge cases where it fails.

---

> > ### Comment · Reviewer_rsmr · 2022-08-09
> > **Response to authors**
> >
> > Thank the authors for the clarifications. The rebuttal addressed my concerns about experimental settings and baseline methods. However, I still feel the online learning paradigm (learning from asking) is crucial in the whole pipeline, as simply choosing between two sets of frozen policies can intuitively face severe problems in new tasks and less expert knowledge. The current contribution of this paper is limited to when to ask, but one intuitive next step is to follow this line of thought and make some analysis on what the frozen policies failed or expected to help the most to provide some hints for future study in embodied AI. Although, as mentioned by reviewer HGan, iterative updates can be easily adopted, the current prompting of frozen policy does not provide much insight into what challenges we might face. Therefore, I'm increasing my score to a borderline, hoping that the authors would clarify more on the potential of the current method as a building block for future studies.

---

> > > ### Author Response · Authors · 2022-08-09
> > > **Author Response to Rebuttal Comments**
> > >
> > > We are happy to hear that our rebuttal has addressed some of your concerns and thank you for increasing your rating of our work.
> > >
> > > **Therefore, I'm increasing my score to a borderline, hoping that the authors would clarify more on the potential of the current method as a building block for future studies.**
> > >
> > > We see two great future directions which can leverage our Ask4Help module to further improve embodied models:
> > >
> > > - Ask4Help can be additionally used as a way of highlighting the limitations of a model, and benchmark how far our off-the-shelf models have come based on the amount of help asked (12% for pre-trained agent v/s 98% for a random agent). For instance, in rearrangement, the Ask4Help policy asks for expert supervision for navigation v.s. interaction actions with ratio of 6:1, this suggests that navigation actions may be more uncertain than interaction actions. This idea is quite general: if we identify some set of potentially relevant features that we believe may contribute to model failure (e.g. mis-recognition, exploration failure, etc) we can use simple statistical models to determine how these features relate to cases where the Ask4Help model queries the expert. For instance, this [plot here](https://anonymous-neurips22.s3.us-west-2.amazonaws.com/a4h/object_type_help_analysis.png) shows amount of help needed per object category, which can tell us which object types are mis-recognized or hard to navigate to. This then provides guidance as to how we can improve existing models.
> > >
> > > - As you and HGan have noted, we can use the expert supervision provided at inference time (perhaps in a federated learning approach) to update the existing model iteratively. Some exploratory experiments of ours suggest that this is not as simple as simply using imitation learning with the given supervision as catastrophic forgetting can occur: continuous learning is a challenging domain with a highly active community that attempts to tackle just these types of problems. We should note that there are some other interesting ideas regarding how one might reuse the queried expert actions (e.g. a semi-parametric memory that stores the expert actions and recalls them when necessary).
> > >
> > > Both of the above directions are interesting problems that, in our view, would likely lead to their own paper. We believe that our paper makes a useful contribution and serves as a good foundation for these future directions.

---

### Official Review · Reviewer_gw2r · 2022-07-10

**Rating:** 8
**Confidence:** 4
**Soundness:** 4 excellent
**Presentation:** 4 excellent
**Contribution:** 4 excellent

**Summary:**

This is a strong paper that examines the question of how to enrich existing policies for embodied tasks (in this case, object navigation and rearrangement) with the ability to “ask for help.” Unlike prior work that defines heuristics for model uncertainty, requires extra supervision in terms of language or subtasks, or requires retraining the base embodied agent policy, the proposed approach — Ask4Help — learns a base policy-agnostic approach for learning when to ask for help, *without the need to retrain the base policy*. Formalized as a separate policy that takes in a set of user preferences, the “belief” (hidden state) of the embodied base policy, and a small number of interactive rollouts in known environments, Ask4Help is able to learn to trade-off task success with “expert load” (amount of queries to the expert) with minimal data, and with remarkable results on two benchmark tasks — RoboTHOR object navigation and AI2-THOR room rearrangement.

This paper further goes above and beyond to show how the proposed system compares to scenarios where users have different preferences (weightings on success rate vs. desired query load), comparisons to ablations with fixed probabilities of picking “expert” actions, comparison of querying actual (vs. synthetic) humans, as well as the robustness of Ask4Help in the presence of noisy experts.

**Questions:**

- How are the reward preferences chosen? Is it right now just a weight on the ultimate success rate?
- Is there a way to regress more versatile/natural preferences from users online?


**Limitations:**

This paper is very transparent about its limitations — namely that this Ask4Help approach only augments an existing policy with the ability to ask for help, not necessarily learn from the provided expert feedback.

I agree that this is probably out of scope for this work, but for future work I’d suggest the authors look into frameworks like Lazy or ThriftyDAgger as ways to bootstrap systems that (1) know when to ask for expert help, and (2) learn/update policies based on that information.


**Strengths And Weaknesses:**

This paper is original and significant, proposing a system that not only makes sense and seems necessary as we build stronger, more powerful embodied systems. The evaluation is incredibly convincing, and the ablations are well chosen and are crucial in showing the efficacy of the proposed approach.

The clarity of the paper is also an added bonus — the motivating examples were clear and helped ground out the early parts of the paper, the approach is simple yet flexible, and in general, led to a strong paper.

The sole weakness is that I wish the user preference component had a little bit more discussion; it’s still not clear to me how this component interacts with the rest of the system (and especially the training/reward function), nor how the various “preference embeddings” are specified/chosen.

---

> ### Author Response · Authors · 2022-08-02
> **Author Response to Reviewer gw2r's comments and questions**
>
> We thank the reviewer for the insightful comments. We appreciate the positive feedback on the originality and significance of the work, convincing evaluations, well-chosen ablations, and clarity of the paper. Discussion of the raised questions follows.
>
>
> -   **Discussion on user preference components and how the embeddings are specified.**
>
>
> We clarify this in the [common response here](https://openreview.net/forum?id=_bqtjfpj8h&noteId=_Ro-f-VRKCe).
>
> -   **How are the reward preferences chosen? Is it right now just a weight on the ultimate success rate?**
>
> During evaluation, as discussed in the [common response here](https://openreview.net/forum?id=_bqtjfpj8h&noteId=_Ro-f-VRKCe), we can provide the user preference as an input. It can be seen as a way of specifying how costly task failure is to the user (or, alternatively, how willing a user is to be bothered with a request for help).
>
> We specify the reward function as a tradeoff between the cost of failure and cost to ask for expert help. It attempts to balance the cost of failure and cost to request expert help to reach a favorable trade-off. During evaluation, as discussed in the [common response here](https://openreview.net/forum?id=_bqtjfpj8h&noteId=_Ro-f-VRKCe), we can provide the user preference as an input.
>
>
> -   **Is there a way to regress more versatile/natural preferences from users online?**
>
> Currently we allow choosing a reward configuration index from 1:30 to specify user preference, like a control knob. One interesting future work could be getting user preferences in the form of natural language.
>
> -   **Suggestions for future work**
>
>
> Thank you for the recommendation on future work. Adapting the underlying policy is a challenging yet interesting problem that many people in the community are working on.
>
> We’re excited to look into these frameworks and see how future works build on our approach to adapt the policies based on expert feedback received.

---

> > ### Comment · Reviewer_gw2r · 2022-08-08
> > **Rebuttal Response**
> >
> > Thanks for addressing my flagged points -- I'll keep my current score! Hoping that my fellow reviewers are also willing to see the merits of this work!

---

> > > ### Author Response · Authors · 2022-08-09
> > > **Author Response to rebuttal comments**
> > >
> > > Thank you for the positive comments and feedback.

---

### Official Review · Reviewer_HGan · 2022-07-11

**Rating:** 6
**Confidence:** 4
**Soundness:** 3 good
**Presentation:** 4 excellent
**Contribution:** 3 good

**Summary:**

This paper is about learning an ask for help (ask4help) policy on top of an already trained embodied agent. Specifically, the ask4help policy first measures the agent's uncertainty at finishing the given task based on its internal belief state using a pretrained Success Prediction Model (SPM). Then, the ask4help policy decides if it should ask for the expert's help (i.e. the next action to do in the environment) or use the pretrained embodied agent's predicted action. In addition, the ask4help can be adapted at inference time to different users' preference regarding the frequency of being "disturbed" by the agent to answer it. Experiments were conducted on two different environments: the RoboTHOR Object Navigation and the Room Rearrangement in iTHOR. Empirically, it was shown that ask4help manages to yield a very higher success rate while requesting the least amount of expert feedback compared to different baselines: the Embodied Agent only, a Naive Helper with predefined frequencies for asking for help, and Model Confusion that measures the agent's uncertainty based on the confidence of the predicted action rather than relying on the agent's belief state.


**Questions:**

- In the RoomR environment, did the authors investigate what type of action (navigation vs. interaction) provided by the expert was the most common?
- What reward configuration is being used during the evaluation?
- What is the distribution of the reward configurations?
- Are the loop-up embeddings for the reward configuration learned?
- Are the authors going to release the code and trained ask4help policy so the community can reproduce the results?

**Limitations:**

The authors did mention two main limitations of the proposed approach. At the moment, expert feedback is not used to improve the embodied agent which can be annoying to a human user. The second limitation of this work is the authors used proxy experts (available for the tested environments) to train the ask4help policy. Using human-in-the-loop for training is not explored in this work (it is still an active area of research).


**Strengths And Weaknesses:**

**What I like about this paper**
- How to reuse existing trained models is an important research direction with positive environmental impact.
- The authors propose a solution to train a single ask4help policy that can deal with different user preferences via sampling different reward configurations during training.

**Potential weaknesses**
- It is not clear to me what reward configuration is used at inference time? Section 4.3.2 and 4.4.2 do mention a **single** reward configuration used for training.
- In 4.4.3, it is said "If more performance models are proposed for the task, the Ask4Help policy will accordingly adjust the amount of expert help requested...". That sentence seems to imply that this will happen at inference time. However, it is my understanding that both the ask4help policy and the Success Prediction Model will need to be retrained to accommodate for the likely different embodied agent's belief $b_t$.


**Originality, quality, clarity, and significance**

This work shares many similarities with active learning but takes place in interactive environments. The proposed approach is a novel combination of existing techniques applied together to tackle the important research problem of deciding when to ask for help. I could see future work building on this to integrate better with human workflow (e.g., using language to ask questions and interpret the answer similar to [Asking for Knowledge, Liu et al., ICML2022]). I found the paper well-written and well-organized. Figure 2 helped me understand the model overall but it was not clear how the Success Prediction Model was getting any training signal since the gradients were blocked. I realized later it is pretrained and frozen. The submission seems technically sound to me except for how the different reward configurations were defined during training.

Overall, I tend to recommend this paper for acceptance because of the research problem being addressed and the proposed solution that doesn't require training an embodied agent from scratch. I might have missed some flaws, especially with respect to active learning.

---

> ### Author Response · Authors · 2022-08-02
> **Author Response to Reviewer HGan's questions and comments**
>
> We thank the reviewer for the insightful feedback. We appreciate the positive comments about the paper that it presents an important research direction with positive impact.
>
>
> We address the questions and concerns below.
>
>
>
> -   **What reward configuration is used at inference time? What is the distribution of the reward configurations?**
>
>
>
>
> For object navigation, in Section 4.3.2, we train the Ask4Help policy with multiple reward configurations as described in [common clarification here](https://openreview.net/forum?id=_bqtjfpj8h&noteId=_Ro-f-VRKCe). Note that the training time for a single reward and multi-reward setting is similar. For inference, we vary the user preference by giving different reward configuration embeddings and generate different Ask4Help behaviors with a single policy. Specifically, for results in Figure 3, we use reward configurations corresponding to $r_{fail} \in \\{-5,-7,-11,-13,-19,-21,-23,-30\\}$.
>
>
>
> -   **In 4.4.3, it is said "If more performant models are proposed for the task, the Ask4Help policy will accordingly adjust the amount of expert help requested...".**
>
>
>
>
> We apologize for the confusion, your understanding is correct. The success prediction module and Ask4help policy would require to be retrained. What we intended to convey was, depending on how good/bad the new model is, the Ask4Help policy will accordingly converge on a favorable trade-off for expert help and task performance.
>
>
>
> -   **In the RoomR environment, did the authors investigate what type of action (navigation vs. interaction) provided by the expert was the most common?**
>
>
> On the validation scenes, we find that the expert provides more navigation actions than interaction actions with a ratio of 6:1.
>
> -   **Are the embeddings for the reward configuration learned?**
>
>
> Yes, they are trained as described in the [common clarification here](https://openreview.net/forum?id=_bqtjfpj8h&noteId=_Ro-f-VRKCe). This training is what enables it to adapt to different user preferences at test-time.
>
>
>
> -   **Are the authors going to release the code and trained ask4help policy so the community can reproduce the results?**
>
>
> The code to reproduce all the presented results is attached in the supplementary material. We plan to do a public release of our full code and models to the community at the end of the anonymity period.
>
> -   **Use of proxy experts to train the Ask4Help policy**
>
>
> As you note, human-in-the-loop training is an exciting and active area of research which we do not attempt to tackle in this work. That said, we were quite happy to see, recall our human expert evaluation Table 1, that our approach can generalize well to the use of human experts at inference time despite being trained with proxy experts.

---

> > ### Comment · Reviewer_HGan · 2022-08-08
> > **Response to authors' rebuttal**
> >
> > Thank you for the rebuttal and for addressing my concerns.
> >
> > After reading the other reviews, I found that reviewer bRWC raised valuable points, especially regarding quantitive/qualitative results. Thank you for rerunning some experiments with the suggested baselines/ablations.
> >
> > > On the validation scenes, we find that the expert provides more navigation actions than interaction actions with a ratio of 6:1.
> >
> > While it's not surprising that navigation actions are more uncertain, I think it speaks more about the limitation of the pretrained agent. I like how Ask4Help policy can be used to highlight a pretrained agent's limitations.
> >
> > Briefly, regarding one of reviewer rsmr's comments:
> > > proposed model possesses the disadvantage that it can not learn from the expert's feedback
> >
> > Not everything has to be end-to-end learning, research that promotes models' reusability is very important. Additionally, iterative learning could be a viable strategy here. For instance, while being deployed in production, an Ask4Help system can still collect data examples at the boundary of the frozen agent policy's capabilities. Then, update the frozen agent policy and redeploy.

---

> > > ### Author Response · Authors · 2022-08-09
> > > **Author Response to rebuttal comments**
> > >
> > > We thank the reviewer for their comments on our response.
> > >
> > > -  **I like how the Ask4Help policy can be used to highlight a pre-trained agent's limitations.**
> > > That is a great point and we thank the reviewer for highlighting this. We would certainly include this in the main draft and it will hopefully allow users to get a deeper insight into their pre-trained agent and tasks.
> > >
> > >
> > >
> > > - **an Ask4Help system can still collect data examples at the boundary of the frozen agent policy's capabilities.**
> > > Thanks for this pointer, and we agree collecting data that can help identify failure mode, then eventually fine-tuning the deployed policy is an interesting direction, and as suggested, is one of the positive merits of Ask4Help.

---

### Official Review · Reviewer_uya8 · 2022-07-12

**Rating:** 4
**Confidence:** 4
**Soundness:** 2 fair
**Presentation:** 4 excellent
**Contribution:** 2 fair

**Summary:**

This paper proposed a method to decide when to ask for expert’s help to improve the task performance of embodied agents. Instead of using heuristics, e.g. model confusion, or expanding the action space of the embodied agent, this paper learns a separate policy on top of the decisions of off-the-shelf embodied agents using reinforcement learning.   The experiments on RoboTHOR show that the proposed method can achieve higher success rates using fewer expert queries.

**Questions:**

- While the paper claims that it can learn efficiently with a fraction of training data, the experiment setup still suggests that it takes 25% of the training scenes to train the Ask4Help model. This is not a small fraction. It will be helpful if the authors can clarify on data efficiency vs. the performance of the Ask4Help model.
- How do different reward configurations affect the number of questions asked? If the reward configuration represents user’s preferences for failure, a simple model can be thresholding the predicted success rate based on the selected user preference and have similar results.

**Limitations:**

This paper has discussed the main limitation of the proposed method which provides opportunities for future work. Another limitation of the proposed method is the fixed user preference, the method doesn’t adapt the number of questions asked based on the interactions with experts.

**Strengths And Weaknesses:**

Strength:
- The proposed method is generic. Without modifying the trained embodied agent, the method can learn when to perform expert queries.
- The proposed method reduces the number of expert queries compared to baselines such as model confusion and naive helper.
- The presentation of the paper is clear. It is easy to follow the paper.

Weakness:
- Missing ablations on different key components. There are two major inputs for the Ask4Help model, success rate prediction and the embodied agent’s belief, it is unclear the contribution of each component. The success rate is a strong indicator of whether the embodied agent needs help. It is likely that a classifier based on success rate is sufficient.
- Unclear how to adapt to different reward preferences at inference time. Ask4Help trains the model with a sample of different reward configurations, but the paper doesn’t show how to estimate the expert’s reward configuration at inference time. Do experts select which reward profile they want? It is also possible that an expert adjusts their reward preference while interacting with the system.

---

> ### Author Response · Authors · 2022-08-02
> **Author Response to Reviewer uya8's comments and questions**
>
> We thank the reviewer for the insightful feedback. We appreciate the positive comments about the method being generic and outperforming the baselines, and the clarity of the presentation. We will now respond to the highlighted questions and concerns.
>
>
>
> -   **Missing ablation on key components.**
>
> -   **The success rate is a strong indicator of whether the embodied agent needs help. It is likely that a classifier based on success rate is sufficient.**
>
>
>
>
> We appreciate the reviewer’s comment. We ran an ablation and observed some interesting results.
>
>
> For the task of Object Navigation, given its simple nature, the success prediction network is able to capture the characteristics of failures reasonably well. We find that by masking out either the belief input or the success prediction input (but not both), we achieve similar results to the ones presented in the paper. Although, we would like to point out that the success prediction uses belief as an input.
>
>
> However, when we try masking out the belief input for Rearrangement, we find that the Ask4Help policy simply learns to request expert help 97% of the time, which is very undesirable from the user perspective. This is due to the complex nature of rearrangement which makes it much harder for success prediction to capture the modes of failure where the agent might require assistance.
>
>
> We believe this indicates that having the agent belief as an input allows our method to be generally applicable to tasks of varying difficulty, especially for complex ones like rearrangement where purely relying on success prediction may not be enough.
>
>
>
> We’ll include this discussion in the updated draft.
>
>
> -   **Unclear how to adapt to different reward preferences at inference time.**
>
>
> We clarify how we train with multiple reward configurations in the above common response [LINK to the common response].
>
>
> -   **Ask4Help trains the model with a sample of different reward configurations, but the paper doesn’t show how to estimate the expert’s reward configuration at inference time. Do experts select which reward profile they want?**
>
>
>
>
> The expert indeed selects the profile that they prefer and this can be easily adjusted at inference time if the expert finds the system is asking for too much (or too little) help. As for the last question, further details can be found at [common clarification comment here](https://openreview.net/forum?id=_bqtjfpj8h&noteId=_Ro-f-VRKCe).
>
>
>
>
> -   **It will be helpful if the authors can clarify on data efficiency vs. the performance of the Ask4Help model.**
>
>
>
>
> Yes, we use 25% of the training scenes for Ask4Help training. We present ablation over how the performance varies when we use just 10% of the training scenes. We train two Ask4Help policies, one with 25% of training scenes like we present in the draft, and one with 10% training scenes. We use the following reward configuration for both cases, $r_{fail} = -10$,  $r_{init\\\_ask} = -1$ and $r_{step\\\_ask} = -0.01$. EP represents expert proportion, an indicator of the amount of expert help is used. The results are as follows:
>
>
>
> | Model setting               | SR    | SPL   | EP (%) |
> |-----------------------------|-------|-------|--------|
> | Ask4Help (25% train scenes) | 86.3  | 33.2  | 12.31  |
> | Ask4Help (10% train scenes) | 84.44 | 32.74 | 12.1   |
>
>
>
> As these results suggest, Ask4Help can learn a fairly good policy with even 10% of the training scenes, which allows us to use it for tasks where such training data might be scarce.
>
>
>
> -   **If the reward configuration represents the user's preferences for failure, a simple model can threshold the predicted success rate based on the selected user preference and have similar results.**
>
>
>
> As we clarify in the “**Missing ablation on key components**” response, the success prediction module works reasonably well for simpler tasks but fails for complex tasks.
>
>
>
> Additionally, picking the right threshold for the given user preference requires access to the validation set to run a hyperparameter search, whereas learning this correspondence as we describe in the [common clarification comment here](https://openreview.net/forum?id=_bqtjfpj8h&noteId=_Ro-f-VRKCe). directly generalizes well on previously unseen scenes. Please see [common clarification comment here](https://openreview.net/forum?id=_bqtjfpj8h&noteId=_Ro-f-VRKCe). for clarification on how this is learnt.

---

> > ### Comment · Reviewer_uya8 · 2022-08-06
> > **Thanks for the response!**
> >
> > I would like to thank the authors for adding the ablations and the clarification of the reward profiles.
> >
> > - It is good to see success rate prediction is not enough for the rearrangement task, but it is possible that a task progress detector can be enough for the rearrangement task given the nature of the rearrangement task is to put multiple objects instead of one object. Also, since it doesn't show any difference when masking out success rate and the belief in object navigation, it seems that we don't need the success rate predictor at all. Only the belief is sufficient to decide if the model should ask questions.
> >
> > - Based on the clarification of the reward profile, the embedding is a lookup table for mapping the selected reward profile to whether to ask a question. If a reward profile is not in the training set, the model cannot handle it. Given the ablation result in the object navigation, one solution is to train a task progress regressor and have different thresholds based on the reward profile.
> >
> > Given these concerns, I would keep my rating.

---

> > > ### Author Response · Authors · 2022-08-09
> > > **Author Response to Rebuttal Comments**
> > >
> > > - **It is possible that a task progress detector can be enough for the rearrangement task given the nature of the rearrangement task is to put multiple objects instead of one object.**
> > >
> > > It is not clear that designing a progress monitoring module for rearrangement is an easier problem than designing an Ask4Help module. Indeed an Ask4Help module needs to only detect one event, that the agent has reached a state where it cannot complete the task alone, while a task progress monitor must have a holistic understanding of the agent's abilities and the environment state throughout the episode. Given the present stream of work, we believe our contribution is a useful one that allows us to reuse existing Embodied AI models.
> > >
> > > Additionally, it is important to note that [rearrangement](https://ai2thor.allenai.org/rearrangement/) is a very complex task that just does not comprise object navigation to multiple objects. It involves interacting with objects, which also includes understanding object affordances and the underlying physics. Additionally, the order in which objects are interacted with is very important since object locations can conflict with each other.
> > >
> > > We would also like to note that, Ask4Help can also give insight on the failure modes of a task. Building on this insight the amount of help asked can be an indicator on how far our existing off-the-shelf have come in terms of autonomous applications.
> > >
> > > - **Given the ablation result in the object navigation, one solution is to train a task progress regressor and have different thresholds based on the reward profile.**
> > >
> > >
> > > We would like to clarify that, even for object navigation, setting the threshold of task progress regressor according to a particular user preference would require access to the unseen validation scenes, however our method generalizes to unseen scenes without ever seeing them. Additionally, this solution would not generalize to tasks of increasing complexity where predicting task success at every timestep is non-trivial.
> > > Additionally, as discussed above regarding progress regression in rearrangement, this solution may be strictly more challenging to implement than our proposed method.
> > >
> > >
> > >  - **If a reward profile is not in the training set, the model cannot handle it.**
> > >
> > > Yes the reward embedding is based on a lookup table, note that we do not claim that it generalizes out of the defined reward profiles. However, the learned policy transfers seamlessly to unseen validation scenes and is able to produce varying behaviors when we vary the reward profiles. We provide options from -1 to -30 which covers a broad range of user preference from almost no help to very liberal help. The 30 reward profiles consider a very wide spectrum of behaviors with performances ranging from ~65% to 93%.
> > >
> > > Additionally, it is also possible to train Ask4Help with more reward profiles if the current set of 30 options are not enough. Generalizing to reward profiles outside the defined ones is something we did not experiment with, since from a user perspective it makes sense to have all possible range of options, rather than trying to generalize for out-of-distribution reward profiles.
> > >
> > > We thank the reviewer for their comment about success prediction. We agree that belief is sufficient to learn a good Ask4Help policy and would update our draft to reflect that. However, our core framework of learning a task-agnostic policy to provide expert help is still applicable and useful for future models and tasks.

---

### Author Response · Authors · 2022-08-02
**Clarification on how to adapt to different user preferences at test time.**

As suggested by Reviewers **uya8, HGan, gw2r and bRWC**, we provide some clarification regarding how we adapt to different user preferences at test time.


To reiterate our reward configuration is

$r_t = r_{fail} + 1_{ask} \cdot r_{init\\\_ask} + r_{step\\\_ask}$ as mentioned in Section 4.3.2.


For instance, one configuration that we use for training is $r_{fail} = -10, r_{init\\\_ask} = -1$ and $r_{step\\\_ask} = -0.01$. Let's refer to this configuration as $R_{-10}$.


Now, as mentioned at the end of Section 4.3.2, if we wish to generate multiple reward configurations we can vary $r_{fail}$ to different values and generate different reward configurations.



Specifically, if we set $r_{fail} = -1$, which would imply that the cost of failure is not very high for the user, we get a reward configuration $R_{-1}$ with $r_{fail} = -1$, $r_{init\\\_ask} = -1$, and $r_{step\\\_ask} = -0.01$.


Following the same trend, we vary, $r_{fail}$ from -1 to -30 to cover a broad range of user preferences and generate rewards configuration namely $R_{-1}$ to $R_{-30}$.


To train an agent with all reward configurations $\{R_{-1}, \ldots, R_{-30}\}$, we uniformly sample a reward configuration $R_{i}$ for each episode in the environment. The reward the agent receives during that episode is based on $R_{i}$.


During validation, we do not have access to these rewards, therefore we need to learn a correspondence between user preference and agent behavior.

To accomplish that, we associate each reward configuration with an index (e.g. $-1 \to R_{-1} \to r_{fail} = -1$), and embed these configuration indices using a standard lookup in a learnable embedding matrix. We provide this embedding as an input to the agent. This allows the agent to learn a correspondence between this embedding input and the cost of failure.

This correspondence is what enables the agent to modify its behavior according to user preference. If the agent is given the embedding corresponding to a high cost of failure then it should request more expert help to assure higher task success.



For results in Figure 3, we train a single policy randomly sampling the reward structure from $\{R_{-1},\ldots, R_{-30}\}$. During inference we simply give different reward configuration indices ranging from $\{-1,\ldots, -30\}$ and the learnt Ask4Help Policy adjusts its behavior accordingly.

We’ll include this discussion in the updated draft to ensure further clarity, thanks to multiple reviewers for suggesting this.

---

### Meta-Review · Area_Chair_W6pK · 2022-08-25

**Recommendation:** Accept
**Confidence:** Certain

**Metareview:**

I thank the authors for their submission and active participation in the discussions. This paper introduces a method for learning a policy that can ask an expert for help, i.e., to obtain the expert action. On the positive side, reviewers found the method to be general [uya8], original and significant [gw2r], intruiging in terms of being able to reuse an existing policy [HGan], and tackling an important problem [rsmr,bRWC], and the paper to be clear [uya8,gw2r,rsmr,bRWC]. In terms of negative points, reviewers were concerned about the novelty [bRWC], unimpressive qualitative results despite strong quantitative results [bRWC], and issues with the range of baselines [bRWC,rsmr] and ablations considered [uya8]. Overall, the paper is borderline. However, bRWC indicated they would raise their score but I don't see this being reflected. Furthermore, in my view reviewer rsmr's concerns regarding baselines and ablations has been addressed by the author rebuttal. Thus, I am siding with reviewers gw2r and HGan, and recommend acceptance. However, I very strongly encourage the authors to further improve their paper based on the reviewer feedback, in particular the points raised by reviewer bRWC regarding the importance of the Success Prediction component of the method.

**Award:**

No

---

### Decision · Program_Chairs · 2022-09-14

Accept